# Antiviral Activities of Algal-Based Sulfated Polysaccharides

**DOI:** 10.3390/molecules27041178

**Published:** 2022-02-09

**Authors:** Jonathan Ardhianto Panggabean, Sya’ban Putra Adiguna, Siti Irma Rahmawati, Peni Ahmadi, Elmi Nurhaidah Zainuddin, Asep Bayu, Masteria Yunovilsa Putra

**Affiliations:** 1Department of Chemistry, Faculty of Mathematics and Natural Sciences, Universitas Gadjah Mada, Bulaksumur, Yogyakarta 55281, Indonesia; jonathanpanggabean@mail.ugm.ac.id (J.A.P.); syaban.putra@mail.ugm.ac.id (S.P.A.); 2Research Center for Biotechnology, Research Organization for Life Sciences, National Research and Innovation Agency (BRIN), Jalan Raya Jakarta-Bogor KM. 46, Cibinong 16911, Indonesia; siti.irma.rahmawati@brin.go.id; 3Faculty of Marine Science and Fisheries, Hasanuddin University, Makassar 90245, Indonesia; 4Center of Excellent for Development and Utilization of Seaweed, Hasanuddin University, Makassar 90245, Indonesia

**Keywords:** sulfated polysaccharides, macroalgae, microalgae, antiviral agents

## Abstract

An antiviral agent is urgently needed based on the high probability of the emergence and re-emergence of future viral disease, highlighted by the recent global COVID-19 pandemic. The emergence may be seen in the discovery of the Alpha, Beta, Gamma, Delta, and recently discovered Omicron variants of SARS-CoV-2. The need for strategies besides testing and isolation, social distancing, and vaccine development is clear. One of the strategies includes searching for an antiviral agent that provides effective results without toxicity, which is well-presented by significant results for carrageenan nasal spray in providing efficacy against human coronavirus-infected patients. As the primary producer of sulfated polysaccharides, marine plants, including macro- and microalgae, offer versatility in culture, production, and post-isolation development in obtaining the needed antiviral agent. Therefore, this review will describe an attempt to highlight the search for practical and safe antiviral agents from algal-based sulfated polysaccharides and to unveil their features for future development.

## 1. Introduction

In early 2020, the World Health Organization announced COVID-19 as a pandemic caused by the alarming and severe damage of the virus’s spread [1]. The virus is identified as a novel betacoronavirus, viz. severe acute respiratory syndrome coronavirus 2 (SARS-CoV-2), which was first identified in China in late 2019 [2,3]. The emergence of a few new variants—including the Alpha variant in the United Kingdom (B.1.1.7), the Beta variant (B.1.351) in South Africa [4], the Gamma variant in Japan and Brazil (B.1.1.28.1), and the Delta variant (B.1.617.2) in India [5], as well as the newly discovered Omicron variant in South Africa (B.1.1.529) [6]—has motivated many efforts to treat and prevent the disease from further casualties. The emerging variants have highlighted the high probability of the emergence or re-emergence of viral infection and test our readiness to fight against them and future unknown pathogens [7]. The readiness may be underlined by a broad research base, including a race to obtain antiviral agents from natural sources.

Recently, sulfated polysaccharides have attracted much interest due to their potential antiviral activities against various strains through various mechanisms of action. They are reported to be active against herpes simplex virus 1 and 2 (HSV-1 and 2) [8,9], dengue virus 2 (DENV-2) [10], a wide range of human rhinovirus (HRV) strains [11], and the H1N1 influenza virus (PR8) or H3N2 [12], human immunodeficiency virus (HIV) [13], DNA damage induced by human papillomavirus (HPV) [14], respiratory syncytial virus (RSV) types A and B [15], viral hemorrhagic septicemia virus (VHSV) of salmonid fish, African swine fever virus (ASFV) [16], and the Varicella zoster virus (VZV) [17]. Recently, sulfated polysaccharides isolated from macroalgae (seaweeds) (*i.e.*, ι-carrageenan) showed antiviral activity against SARS-CoV-2 from an in vitro assay [18] in a promising (but limited) clinical trial [19].

Polysaccharides embedded with sulfate groups are known for their utilization in the foods, nutraceuticals, and cosmeceuticals industry as antioxidant, anticoagulant, and cholesterol-lowering agents [20,21]. These macromolecules can be obtained from marine algae and several bacteria, but they are not found in terrestrial plants [22]. The former source has attracted much attention since algae have a fast growth rate, produce multiple valuable chemicals, and act as natural CO_2_ sequesters. Owing to these characteristics, their utilization is attractive, as is their integration with other valuable chemicals in reducing CO_2_. Furthermore, sulfated polysaccharides derived from algae are obtained in relatively large quantities, with up to a 76% yield, and their extraction process can be performed by simple extractions [23]. In addition, algal-based sulfated polysaccharides are generally safe because many of them have been utilized in the foods and cosmeceuticals industry. Some of them have been recognized by the US Food and Drug Administration to be utilized for human consumption.

This review will highlight the opportunity to further explore the utilization of algal-based sulfated polysaccharides as producers of antiviral agents and will emphasize one of the strategies to overcome the future emergence or re-emergence of pathogens.

## 2. Macroalgae and Microalgae: An Overview

### 2.1. Macroalgae

Macroalgae or seaweeds are members of a broad group of algae with a distinctive macroscopic size. These colorful plants can be found in various aquatic climates, from polar to tropical marine environments. The apparent features on their photosynthetic pigment have led macroalgae to be categorized as green algae (Chlorophyta), red algae (Rhodophyta), and brown algae (Phaeophyta) [24]. The diversity of macroalgae has been known and utilized to benefit humans for many centuries. It has reached its development as an industry worth USD seven billion based on various geographical features of the marine environment from the coastline of Ireland to South East Asia. The practicality of macroalgae is clear from their industrial development as food sources and their ability to enhance other aspects of human life, from biofuel to medicine [25,26].

Macroalgae contain low levels of lipids (0–3%), medium amounts of protein (10–47%), and up to 60% carbohydrates diversified among various species (Table 1) [27,28]. As one of the main constituents of macroalgae, the carbohydrate content has been extensively studied, both in terms of its utility and its production process, especially the content of sulfated polysaccharides such as ulvan from green algae, fucoidan from brown algae, and carrageenan from red algae [29]. Carrageenan is well-known for its utilization as an antioxidant and gelling agent in the cosmeceuticals industry, providing an enhanced consistency and texture [30]. On the other hand, fucoidan has been recognized for its bioactivities in enhancing the melanoma inhibition of lapatinib, anti-fibrotic effects, and anti-inflammatory and antioxidative effects [31,32]. In turn, like other sulfated polysaccharides, ulvan also exhibits the potential to induce intestinal cytokine production as an immunostimulatory agent and to protect liver damage due to oxidative stress [33,34].

The production aspects also brought extensive attention to the recent development of providing valuable products embedded on macroalgae. Recently, Bahari and colleagues conducted an alkali-free extraction of carrageenan, resulting in the linear correlation between a carrageenan-rich precipitation yield and the combination of the temperature–time parameter. Despite the need to further study the exact composition of precipitated solids, these findings suggest that a more environmentally feasible extraction process can be developed [42]. On the broader scale of macroalgae, the production of nutritional composition is affected by the seasonal factor, seawater temperature, nutrient availability, salinity, and light exposure [39].The seasonal factor has been described by Mansilla and colleagues, who showed a negative correlation between the protein and carbohydrates composition of a brown alga extract, *Macrocystis pyrifera*. The highest carbohydrate content obtained from the winter and spring seasonal analysis was associated with the highest amount of protein [43]. This finding was also reported by Garcia-Vaquero and colleagues, who found a negative correlation between the protein and carbohydrate content and a positive correlation between light exposure and the production of antioxidant compounds during periods of increased oxidative stress damage in the summer and spring [44].

### 2.2. Microalgae

On the other hand, microalgae have a distinctive micro-scale cell size but possess the same ability to perform a photosynthetic process, which is commonly perceived to include “blue-green algae” (i.e., cyanobacteria). In comparison to macroalgae, microalgae have been harvested and utilized on a large scale to produce fatty acids, such as docosahexaenoic acid (DHA), and feed for aquacultures owing to their potential protein content [45,46]. This unicellular organism is well-known for its high protein composition, which relies on various aspects of cultivation and species (Table 2) [47]. Cultivation conditions, such as pH and salinity, alter growth rates alongside the protein, carbohydrate, and lipid content [48]. The nitrogen concentration in the culture medium also induces the production of protein, and the combination of high nitrogen and low temperatures has been found to increase the content of ω-3 fatty acids, such as α-linolenic acid (ALA) and docosahexaenoic acid (DHA) [49]. The carbohydrate content of microalgae has also been studied as one of the parameters of culture optimization; however, its bioactivities require further attention. Recently, Kumaran and colleagues optimized the culture condition for *Nannochloropsis oceanica*, resulting in an almost 45% increase in the carbohydrate content, which may provide insights into the up-scalability of production [50,51].

In summary of the above discussion on the proximate content of macroalgae and microalgae, carbohydrates are in the lead among other main constituents compared to the protein, fat, and ash content. The carbohydrate content, including polysaccharides, from marine algae correlates with the presence of sulfated polysaccharides. A study on *Gymnogongrus torulosus* polysaccharides extraction shows that, of the total polysaccharide content obtained, 35.6% is of the carrageenan type [56]. This study is supported by another study on *Gymnogongrus tenuis*, which was reported to provide 90% of the carrageenan type from the total polysaccharide obtained [57]. The study on galactans from *Kappaphycus alvarezii* also found a major constituent of κ- and µ-carrageenan (~74% and ~3%, respectively) from the total obtained polysaccharide. This study also includes the possibility of finding an agaran-type sulfated DL-hybrid of galactans at a significant level (~14%) from the total extract [58]. These studies indicate the potential to obtain sulfated polysaccharides as a member of carbohydrates due to their abundancy and potent biological activities.

## 3. Algal-Based Sulfated Polysaccharides

Sulfated polysaccharides are a diverse group of anionic polymers. In algae, these macromolecules have critical supportive and protective functions for the cell walls of algae, thus promoting the high ionic strength of marine environments, pathogens, and moisture control [22]. These features allow them to carry out various biological activities, including antiviral activities.

### 3.1. Carrageenan

Carrageenan is commonly extracted from the various genus of red macroalgae, such as Chondrus, Eucheuma, Gigartina, and Hypnea. Its first isolation was extracted from *Chondrus crispus* or Irish moss [59,60]. It is isolated through the hot alkaline extraction of the biomass. Its structure contains repeating galactose and 3,6-anhydrogalactose units linked by alternating α-1-3 and β-1-4 glycosidic linkages [61]. In general, carrageenan is categorized into various common types: κ-(kappa), ι-(iota), ι-(lambda), µ-(mu), and ν-(nu) carrageenan. The first three types are characterized according to their inclusion of one, two, and three sulfate ester groups on each dimer, respectively (Figure 1) [62]. The other types (µ- and ν-carrageenan) are considered as precursors of κ- or ι-carrageenan through the formation of a 3,6-anhydrogalactose bridge under alkaline conditions [63]. Alkaline extraction provides a crude carrageenan, which requires a refining process to obtain high-quality carrageenan. The purification process involves a few techniques, such as freeze-thawing, which includes the gelation of crude extract prior to freeze-drying and alcohol precipitation using 2-propanol or another alcohol. The selection of this process depends on the relationship between the physical properties embedded and the chemical structure of the carrageenans. Freeze-thawing is used to obtain κ-carrageenan, whereas ι- and λ-carrageenan powder are attained via alcohol precipitation [21].

Carrageenan is hydro-soluble, but its solubility depends on the amount of the ester sulfate group within the structures. λ-carrageenan is the most soluble species in water, followed by ι- and κ-carrageenan. Due to the presence of ester sulfate, the solubility of carrageenan can be altered by the presence of cations. Sodium salts of ι- and κ-carrageenan can improve their solubility in cold water. Meanwhile, the potassium salts of ι- and κ-carrageenan are soluble only in hot water [64]. Moreover, the addition of potassium ions can produce carrageenan aerogels with different porosities, including meso- and microporous, which can be tuned and utilized in extended-release drug applications. Furthermore, carrageenan generally has a viscosity of more than 5 cPs at 75 °C for a 1.5% solution, which decreases below a pH of 4.3 and lowers the gel strength by 20–25% for each 10 °C increase in the process temperature and for each 0.5 pH unit decrease [65].

### 3.2. Agaran

Similar to carrageenan, agarans are a polysaccharide that can be found on algae. They have a similar structure as carrageenan with broader functional groups, such as sulfate, methyl, or pyruvate, instead of the exclusive sulfated polysaccharides [66]. In contrast with carrageenan, agarans are commonly found in two fractions (agarose and agaropection), which are different regarding their ionic charges, where the neutral fraction of agar (known as agarose and agaropectin) serves as the charged fraction. Agarans may be identified by ^13^C-NMR spectroscopy to allow L-galactose to be identified as the main constituent [62]. However, this identification may be affected if both polysaccharides exist in a hybrid form, as isolated by Zuniga and colleagues from *Schizymenia binderi*, who showed the unique position of the sulfate attachment located on the *O*-3 position of α-L-galactopyranosyl, as well as the glycosylated *O*-2 position [67]. Despite the existence of this hybrid form, the agaran may be found as a significant component in room temperature water-extracted macroalgae *Gymnogongrus tenuis*, accounting for slightly more than 60% of its composition (Figure 2) [57]. Agaran can be extracted via hot water extraction above their gel melting points (~85 °C). It is used as a gelling agent in food industries, as the gel maintains its stability and viscosity within a wide pH range of 4.5 to 9.0 [64,68].

### 3.3. Fucoidan

Fucoidan is a sulfated polysaccharide constituent of brown macroalgae with fucopyranose as one of its building blocks. It is typically obtained from brown macroalgae genera of Fucus, with up to a 62% yield. It is a water-soluble polysaccharide with a backbone consisting of α-(1→3)-linked L-fucopyranose or an alternating pattern of L-fucopyranosyls linked at the α-(1→3) or α-(1→4) position (Figure 3). Different substituents could also be attached to the fucoidans’ backbone (e.g., acetate, sulfate, or other glycosyl units) [69,70]. The wide variety of these substituents makes it difficult to extract fucoidan to obtain high-quality fucoidan due to a few challenges. Separation could be carried out by utilizing the sulfate or other anionic properties of the fucoidans’ substituent. For instance, these anionic groups could be exchanged with the negative ions contained in anionic resins, and the impurities could be eluted by an appropriate solvent, such as sodium chloride [71]. Another approach for the fucoidans’ purification is shown by Zayed and colleagues, who utilized and immobilized a polycationic perylene diimide derivative to obtain a high-quality grade fucoidan with 97% purity [72].

### 3.4. Porphyran

Porphyran comes from a specific genus of red macroalga of Porphyra. These polysaccharides’ backbone consists of alternating 1,4-linked α-L-galactosyl-6-sulfate and 1,3-linked β-D-galactosyl units, with a small number of 3,6-anhydro-α-L-galactosyl units (Figure 4). A variation, however, can occur in the backbone (indicated by the methylation on the D-galactosyl units at the C-6 position) due to the extraction of *Porphyra capensis* [73,74]. Porphyran may also be obtained from a discolored “nori” *P. yezoensis*. Isaka and colleagues studied the discolored nori as an alternative porphyran source; the best result was 20.6 g of porphyrin per 100 g (compared to 10.6 g per 100 g of normal nori using ethanol maceration and heat treatment) [75]. Due to its high solubility in water, porphyran is commonly extracted by a conventional water maceration method that utilizes mechanical stirring at room temperature. However, applying physical forces, such as microwave-assisted extraction (MAE), resulted in a 191% increase in yield with the conserved functional group of the polysaccharide (from 1.72% using traditional hot water extraction to 5.01% using MAE) [76].

### 3.5. Ulvan

Ulvan is a type of sulfated polysaccharide commonly found in the genus Ulva. The backbone of ulvan consists of the repeating of a disaccharide unit, namely, ulvanobioronic acid, which has two types (type A and type B), and ulvanobioses as the member of type U (Figure 5). Type A ulvan contains (1,4) linked β-D-glucuronic acid with α-L-rhamnose 3-sulfate, whereas type B can replace β-D-glucuronic acid with α-L-iduronic acid. Type U, on the other hand, has β-D-xylose (1,4) linked with the main α-L-rhamnose 3-sulfate [77,78]. Ulvan may be isolated through extraction via a salt removal process followed by acid extraction using HCl. This acidic process is deemed superior to sodium oxalate extraction, owing to its higher yield (8.2% vs. 4.0%). In addition, the acid extraction method provides selective extraction products with fewer proteins (0.4–0.7%) than sodium oxalate (4.1–5.9%) [79], which indicates the significant impact of pH on the extraction of ulvan.

### 3.6. Exopolysaccharides

Exopolysaccharides are a specific type of polysaccharide defined as an extracellular polymeric substance. Exopolysaccharides are divided into two main groups: cell surface-associated exopolysaccharides, and those released into the surrounding environment. The association with the cell surface may be referred to as a sheath, which resembles a thin and dense layer in the surrounding cell, or capsules with layers of exopolysaccharides that resemble a cell shape. Another association with the cell surface is called slime; in this case, exopolysaccharides are dispersed around the organism without resembling a cell shape (Figure 6) [80]. The chemical composition of exopolysaccharides is diverse and different from other polysaccharides with sugar/monosaccharide members from the pentose, hexose, uronic acid, and amino sugar groups. Other constituents found on exopolysaccharides are non-sugar members, such as sulfuric acid and phosphoric acid (which are attached by ester linkages), acetic acid (which is attached by *O*-acyl linkages and *N*-linkages), succinic acid (which is attached by *O*-linkages), and pyruvic acid (which is attached by acetal linkages; Figure 7) [81]. Recently, Gaignard and colleagues screened 66 microalgae and cyanobacteria and found 17 new monosaccharides as a constituent of the secreted exopolysaccharides that are related to different phylogenetic affiliations [82].

## 4. Antiviral Activities of Algal-Based Sulfated Polysaccharides

The wide variety of structural features and physical properties, along with the advancement of the downstream process in providing sulfated polysaccharides derived from algae, has been investigated in the study, which involves an assessment of their biological activities. A few algal-based sulfated polysaccharides are already well-known for their application as food additives and in the cosmeceutical industry by utilizing their antioxidative, anti-inflammatory, and anticoagulative properties, along with their potency to inhibit adipogenesis [23,63,83]. As a result of their multiple biological activities and their safety, they have gained attention as an antiviral agent [84,85,86]. In particular, such attention is driven by the urge to obtain a suitable agent to prevent further casualties of respiratory infection caused by the SARS-CoV-2 strain and to protect against new and re-emerging viruses [87]. Herein, we attempt to present the potential antiviral activities from sulfated polysaccharides and their relationship with the embedded physicochemical properties segregated by the respective active compound and summarized in Table 3.

### 4.1. Carrageenan

Carrageenan is a sulfated polysaccharide with remarkable attention on its potency as an inhibitor of respiratory virus. A study on ι-carrageenan revealed its superiority for inhibiting the HRV-infected HeLa cell line when compared to λ- and κ-carrageenan. It was found that ι-carrageenan can completely block HRV2-induced cell death, but there is only a 55% cell protection for λ-carrageenan and 62% for κ-carrageenan. It has also shown the potency to inhibit the replication of various strains of HRV, HRV1, 14, 16, 83, and 84, with more than a 99% inhibition at 5 µg/mL. In addition, the study emphasizes the safety of ι-carrageenan, as no toxicity was observed against human nasal epithelial cells (HNep) at >500 µg/mL and against the HeLa cell line at >1000 ug/mL (Table 3) [11].

A recent study by Morokutti-Kurz et al. revealed that ι-carrageenan inhibited the cell entry of SARS-CoV-2 pseudotyped lentivirus in a dose-dependent manner, with a virion particle neutralization value of 2.6 µg/mL of IC_50_ (in ACE2-HEL293 cells/well infected with SARS-CoV-2 spike pseudotyped lentivirus). This finding confirmed that ι-carrageenan interacted with the S glycoprotein of SARS-CoV-2 to inhibit viral infection (Figure 8). Interestingly, it had excellent activity in inhibiting 79% of the SARS-CoV-2 virus at 10 µg/mL compared to λ- and κ-carrageenan, which showed an 80% inhibition activity at 100 µg/mL. In addition, carrageenans did not show any toxicity in Vero B4 cells, as a 50% cell reduction survival was not reached at the highest tested concentration (100 µg/mL). This outcome indicates its safety for medical treatment (Table 3) [88]. In contrast, another study revealed that non-sulfated polysaccharides (carboxymethylcellulose and hydroxypropylmethylcellulose) and low-molecular-weight galactose-4-sulfate were inactive at 100 µg/mL. Chondroitin sulfate, which contains one sulfate group on each dimer, also did not appear to significantly inhibit SARS-CoV-2 on Vero E6 cells [89]. These studies suggest that the activity might be related to the number of sulfate groups and the chain length of the observed polysaccharides.

**Table 3 molecules-27-01178-t003:** Summary of antiviral activity among carrageenans with each respective proposed mechanism of action and remarks on molecular weight and sulfate content of the tested compound.

Sulfated Polysaccharide	Virus Strain	Antiviral Activities	Proposed Mechanism of Action	Toxicity (Cell)	Remarks on Molecular Weight and Sulfate Content	Refs.
	CP (%)					
ι-Carrageenan	HRV2	100% ^2^	ι-Carrageenan inhibits HRV2 entry to infect HeLa cell line	>1000 µg/mL (HeLa)		[11]
κ-Carrageenan		62% ^2^				
λ-Carrageenan		55% ^2^				
	log TCID_50_ ^1^					
ι-Carrageenan	HRV2	<2 ^2^				
κ-Carrageenan		~6 ^2^				
λ-Carrageenan		~6 ^2^				
	Viral Replication Inh. at 5 µg/mL					
ι-Carrageenan	HRV1A	>99%	Reduces production of HRV particles on HeLa cell line	>500 µg/mL (HNep)		[11]
	HRV14	>99%				
	HRV16	>99%				
	HRV83	>99%				
	HRV84	>99%				
	Neutralization Activity					
ιCarrageenan	SARS-CoV-2 Spike pseudotyped lentivirus	79% ^3^	Inhibits cell entry of the SARS-CoV-2 spike pseudotyped lentivirus	>100 µg/mL (Vero B4)	1. High molecular weight Fucoidan from *U. pinnatifida* and *F. vesiculosus* shows less than 50% reduction in infection 2. Polymer without sulfate found to be inactive	[88]
κ-Carrageenan		~80% ^4^				
λ-Carrageenan		~80% ^4^				
	EC_50_ (µg/mL)					
λ-Carrageenan	SARS-CoV-2	0.9	1. Neutralizing viral glycoprotein HA 2. Blocking the VP harboring viral ribonucleoprotein complexes	>300 µg/mL (MDCK)	λ-carrageenan (1025 kDa) shows better solubility in cold water than other carrageenans because of its higher sulfate content	[90]
	Influenza A (H1N1)	0.3				
	Influenza A (H3N2)	0.3				
	Influenza B	1.4				

CP = cell protection; HRV = human rhinoviruses followed with each respective strain number; HeLa = human cervical epithelial carcinoma cell line; HNep = human nasal epithelial cells; Vero B4 = embryonic African green monkey kidney cells; HA = hemagglutinin; VP = viral particles; MDCK = Madin–Darby canine kidney cells; ^1^ value of viral replication inhibition; ^2^ at 200 µg/mL concentration; ^3^ includes 2.6 µg/mL IC_50_ for ι-carrageenan and percent neutralization obtained at 10 µg/mL; ^4^ at 100 µg/mL.

Furthermore, Morokutti-Kurz et al. observed a low activity on fucoidan by only a ~50% inhibition (compared to ι-, λ-, and κ-carrageenan by a <40% inhibition) at 100 µg/mL [88]. Although λ- and κ-carrageenan were active against SARS-CoV-2, the researchers observed a small fraction of ι-carrageenan (27% and 16%) in two carrageenan samples based on NMR analysis. The possible responsibility of ι-carrageenan to enhance the activity of the two latter carrageenans should not be excluded. In the specific case of λ- and κ-carrageenan alone, Jang et al. confirmed the activity of λ-carrageenan against the virus entry of SARS-CoV-2, together with influenza A and B. Herein, λ-carrageenan with a molecular weight of 1025 kDa had an EC_50_ of 0.9 ± 1.1 µg/mL and a CC_50_ of >300.0 µg/mL (selectivity index of >333.3) in the in vitro test of Vero cells infected with SARS-CoV-2, whereas remdesivir had an EC_50_ of 23.5 ± 1.2 µM (~14.16 µg/mL) and a CC_50_ of >300.0 µM (selectivity index of >12.8) [90]. Moreover, its EC_50_ values against influenza A and B viruses were 0.3 to 1.4 µg/mL, with no cytotoxicity up to 300 µg/mL (selectivity index >263.2). The contamination between carrageenan types might be a concern for the sole activity embedded on both the known potency and the selectivity of carrageenans involving a structural feature relationship with their bioactivities [93]. Regardless of the contrast and possible impurities, the study on tuning the potency of carrageenans by the information on the composition and structural feature, such as the molecular weight distribution, number of sulfate esters, and their position, together with bioactivities, could provide further interesting findings. This matter remains to be discovered.

### 4.2. Fucoidan

Owing to it being moderately active in inhibiting SARS-CoV-2 compared to sea cucumber sulfated polysaccharides, fucoidan has drawn considerable attention as a potential antiviral agent. A study on fucoidan isolated from *Adenocystis utricularis* showed that it inhibits HSV-1 and HSV-2 by 1.25 and 1.63 µg/mL of IC_50_, respectively. The IC_50_ value was also found following an amount of fucose by 84%-mol of the extracted product compared with the lowest fucose content by 60%-mol with 4.79 µg/mL and 8.46 µg/mL of IC_50_, respectively (Table 4) [94]. A further study conducted on fucoidan isolated from *Undaria pinnatifida* supported the finding related to the HSV-1 and HSV-2 inhibition potential by 2.5 and 2.6 µg/mL IC_50_, respectively, with the addition of influenza A by 15 µg/mL, where fucoidan was added at the same time as the viral infection. The mechanism of action is proposed by inhibiting virus–host cell-binding. The mechanisms were observed by the decrease in IC_50_ on the addition of fucoidan after 1 h of virus infection by 14, 5.1, and 55 µg/mL IC_50_ against HSV-1, HSV-2, and influenza A virus, respectively (Table 4) [95]. Furthermore, Song et al. reported that ι-carrageenan inhibited only 50–60% SARS-CoV-2 infection on Vero E6 Cells at ≥125 µg/mL [86,89]. Sulfated polysaccharides isolated from sea cucumber *Stichopus japonicus* (SCSP), which is mainly composed of fucosylated chondroitin sulfate and fucoidan, had an IC_50_ of 9.10 µg/mL, and fucoidan itself inhibited 60% of viral infection at ≥15.6 µg/mL. The study also noted that a high level of structural flexibility is required for the polysaccharide to bind to S glycoprotein. The flexibility is highlighted by the tighter binding between the sulfated glycosaminoglycan and envelope protein compared to the more rigid suramin [96]. The flexibility of sugar chains has been reported as one of the main features of polysaccharide solutions’ confirmation, together with morphological characteristics and polysaccharide conformation. The presence of sulfates on a polysaccharide chain also improves the solubility and antioxidant activity by forming a more outstretched conformation than non-sulfated polysaccharides [97,98,99].

Findings related to the influenza A virus were derived from the further assessment regarding fucoidan isolated from *Kjellmaniella crassifolia*, with exceptional results from other studies. The isolated sulfated polysaccharides were found to have inactivation capabilities after being utilized to pretreat a mouse-adapted H1N1 influenza virus (PR8) and H3N2 (a variant of the influenza A virus) by 30.5 and 6.3 µg/mL on the concentration required to inhibit the plaque number by 50%. In addition to its inactivation properties, fucoidan inhibited the influenza A virus neuraminidase (Figure 8), an enzyme that promotes virion entry and the release of the influenza A virus by 8.8 µg/mL of IC_50_. The enzyme inhibition further supported in vivo evidence that fucoidan enhances the PR-8-infected mice’s survival and decreases pulmonary virus titers [12]. Further support for the potency of fucoidan was obtained from a different perspective of the symptoms and remarked treatment of respiratory infection, with its potential to be produced as fucoidan preparation and act as a supplementary agent. This potential is indicated by reduced allergic asthma symptoms via airway inflammatory response attenuation and mucus hypersecretion. In addition, fucoidan can significantly inhibit inflammatory mediators associated with toll-like-receptor-3 (TLR3) when a 1 mg/mL solution is prepared in vitro [100,101,102].

### 4.3. Agaran

For less sulfated polysaccharides, agaran might be accompanied by sulfate constituents, and a few studies have discussed its antiviral activities. A study on sulfated agarans isolated from *Bostrychia montagnei* revealed the antiviral activities against HSV-1 and HSV-2, with the active fraction belonging to the higher sulfate content (17% to 24%) rather than to the lower sulfate content (11.2% to 16.2%). Active fractions can deactivate the other fraction by ≥50 µg/mL of IC_50_ to reduce plaque formation in Vero cells to a greater extent than the large range of 12.9–25.7 µg/mL of IC_50_ for HSV-1 and 11.2–46.2 µg/mL of IC_50_ for HSV-2 (Table 5) [103]. Further studies on the sulfated agaran isolated from *Acanthophora spicifera* yielded the same result, confirming that the dominance of the active fraction against HSV-1 and HSV-2 leads by the fraction with a higher amount of sulfate. These findings suggest the importance of the sulfate group attached to polysaccharide backbones in antiviral activity [104]. They also emphasize that additional valuable information regarding sulfate modification and its antiviral activities can be obtained.

### 4.4. Porphyran

Porphyran has received little attention regarding its potential status as an antiviral agent. One study suggests that a dietary supplement of *Porphyra umbilicalis*, a member of the genus Porphyra known for producing porphyran, can increase hyperplastic epidermal lesions from 36 to 100%, while also inhibiting DNA damage induced by human papillomavirus (HPV) on transgenic mice [14]. A porphyran derivative called oligoporphyran could also retain the potential to provide antiviral activities. It was found that these derivatives have a better antioxidant effect by 2.01 mg/mL IC_50_ of scavenging effect in comparison with a higher molecular weight by 26% inhibition at 1.10 mg/mL. In addition, an enhancement of immunomodulation is also indicated by degraded porphyran at ~130 µmol/L of NO secretion in comparison to a positive control lipopolysaccharide and non-degraded porphyran (~70 and ~100 µmol/L, respectively) [74,107,108].

### 4.5. Ulvan

The bioactivity potential exhibited by ulvan regarding antiviral activities against HSV-1 with 373.0 and 320.9 µg/mL EC_50_ were found from the enzyme-assisted extraction (EAE), with a multiple-mix of glycosyl hydrolases and exo-β-1,3(4)-glucanase of *Ulva armoricana*, respectively (Table 5). The enzyme-assisted extraction also revealed that the product of EAE from the group of carbohydrases is superior to the protease enzyme in terms of antiviral activity against HSV-1 for the inactivity (>500 µg/mL EC_50_) of the product. The difference among enzyme types used also correlates with an increase in rhamnose (the main component of ulvan) in the carbohydrase products. This further implies that the selective manner of the enzyme might enhance ulvan production [105]. A further assessment of the antiviral potential of ulvan is exhibited in one study that reported its ability to inhibit syncytia formation by 51.54% at 0.1 µg/mL without significant virucidal activity against Newcastle disease virus (NDV) virions while exhibiting a low cytotoxicity on Vero (African green monkey kidney) cells by 810 µg/mL IC_50_. The lack of virucidal activity of the isolated ulvan from *Ulva clathrate* might correlate with the inhibition of the NDV replication cycle [92], which also contributes to the known mechanism of the action of sulfated polysaccharides (Figure 8).

As further evidence of ulvan’s potential as an antiviral agent, it was found to inhibit Japanese encephalitis virus (JEV) infection in Vero cells by 70% in terms of cell viability compared to a control condition. A further examination of the extract from *Ulva lactuca* against JEV reveals the ulvan’s ability to form complexes with JEV and, thus, to prevent JEV from entering cells. Other research cites the efficacy of *Ulva lactuca* extract in vivo, which shows a 40% survival rate among JEV-infected mice (there were no survivors in the control group) [106].

### 4.6. Exopolysaccharides

The large group of exopolysaccharides also contributes to the search for antiviral agents from algal-based sulfated polysaccharides. An early potency was found on the marine microalgae *Cochlodinium polykrikoides*. The isolated extracellular sulfated polysaccharides can have a cytopathic effect on influenza virus types A and B, as well as respiratory syncytial virus (RSV) types A and B, and HIV-1. A further fractionation process showed inhibition against the mentioned strains from obtaining two fractions that exhibit different antiviral activities. Fraction A1 was found to inhibit only HSV-1 but not parainfluenza virus type 2; the other fraction, A2, was active in the opposite manner on the viral strain (Table 6), despite their similar monosaccharide composition and IR spectra analysis [15].

Another finding suggests that the aqueous extracts of various marine microalgae, namely, Porphyridium *cruentum*, *Chlorella autotrophica*, and *Ellipsoidon* sp., significantly inhibit viral infection hemorrhagic septicemia virus (VHSV) in salmon fish and the African swine fever virus (ASFV) (Table 6). The mechanism underlying this inhibitive effect correlates with the in vitro replication of both viruses in a dose-dependent manner [16]. These findings have made it possible to study the constituent’s ability to inhibit RNA and DNA [109,110]. They also contribute to the development of broad-spectrum antiviral agents, especially human-related strains [111]. The understanding of human-related antiviral activity is advanced by findings on cell wall sulfated polysaccharides isolated from *Porphyridium* sp. The isolated polysaccharides were shown to inhibit HSV-1, HSV-2, and Varicella zoster virus (VZV) without any cytotoxic effects on the Vero cell line [17].

Another study on exopolysaccharides isolated from *Porphyridium cruentum* also focused on the culture medium influence in the production. A magnesium cation additive (using MgCl_2_ and MgSO_4_) increased the biomass and exopolysaccharides’ yield, leading to slightly better antiviral activity against HSV-1, HSV-2, Vaccinia, and *Vesicular stomatitis* virus (Table 6). The inhibition of HSV-1 is superior compared to the control medium by 56 µg/mL of EC_50_ for the control medium and 34–38 µg/mL of EC_50_ for the magnesium-altered medium, together with the *Vesicular stomatitis* virus, which was inhibited by exopolysaccharide by 100 µg/mL of EC_50_ for the control condition and 20–56 µg/mL of EC_50_ for the magnesium-altered medium. For HSV-2 and the *Vaccinia* virus, the inhibition rate improved after adding MgSO_4_ by 12 µg/mL EC_50_ for both strains and by 10 µg/mL EC_50_ after adding MgCl_2_ addition and applying the control condition for both strains [112].

## 5. Outlook and Future Prospects

Sulfated polysaccharides are well-known for their application in the food and cosmeceuticals industry. These macromolecules are commonly obtained from marine algae and several bacteria, but are not found in terrestrial plants. One survey conducted by Aquino and colleagues revealed that all polysaccharides from marine plants are sulfated [113,114]. This finding is supported by another study on the biosynthesis of sulfated polysaccharides, which found the unshared presence of carbohydrate sulfotransferases and sulfatases on marine algae with terrestrial plants [115,116]. This finding on 16 new different sulfatases from Chlamydomonas and an increased number of the class member exhibited a vast diversity of specificity on the substrate and produced compounds to be discovered [115,117].

Herein, the use of carrageenans is especially attractive since most of them have been recognized by the US Food and Drug Administration as “generally recognized as safe” (GRAS) for human consumption (21 CFR 172.620). They also present a low cytotoxicity from other types of sulfated polysaccharides (see Table 3, Table 4 and Table 5). In addition to their common utilization as a gelling agent in the food industry, their potential broad biological activities have attracted much attention for producing high-quality products for medical treatments. Particularly, the COVID-19 outbreak and emerging variants of the SARS-CoV-2 virus have highlighted the high probability of the emergence or re-emergence of viral infections. They have also tested our readiness to fight against them (and future unknown pathogens) by exploring the potency of antiviral agents from natural sources, including algal-based sulfated polysaccharides.

According to various studies, carrageenan has remarkable potential to be used as an in vitro and in vivo antiviral agent, owing to its wide range of antiviral activities, followed by its relative, fucoidan. Despite the lower amount of attention given to fucoidan, the sulfated polysaccharides could still become antiviral agents. On the other hand, other sulfated polysaccharides have been understudied, providing an opportunity for future researchers.

One research direction that could be followed is the purification of isolated sulfated polysaccharides. The continued study on fucoidan isolated from *Adenocystis utricularis* has shown that further precipitation using cetrimide improves antiviral agents’ activities against HSV-1 and HSV-2 by 0.28 µg/mL and 0.52 µg/mL of IC_50_, respectively. The fractionated fucoidan has distinctive features, remarked by the uniform constituent of galactofucan [94]. This finding indicates that contaminants or impurities in the fraction interfere with its antiviral capabilities. Along with the interference of antiviral capabilities, the information on the composition determines which component of the complex sulfated polysaccharides holds the potency and indicates the probability of discovering a new synergistic effect between the various composition percentages of different constituents.

Post-isolation processing could also improve antiviral activities. The study of the oligosaccharide of κ-carrageenan has revealed its effective inhibition of influenza A H1N1 virus replication. A unique mechanism of action was also observed by a derivative of κ-carrageenan, as it cannot bind to the cell surface of Madin–Darby canine kidney (MDCK) cells, unlike other sulfated polysaccharides. Fortunately, this phenomenon is backed up by the ability to enter the MDCK cell and to subsequently inhibit mRNA and protein expression after the internalization stage of the virus. Not only the ability to inhibit virus replication, but also the study on oligosaccharide of κ-carrageenan, indicate that oligosaccharide with a high molecular weight does not inhibit influenza A H1N1 virus replication at a high rate. The relationship between antiviral activity and the molecular weight might be correlated with water solubility and a cell-penetrating ability [118,119]. The proposed strategy to use a higher molecular weight is also supported by recent studies conducted by Kwon and colleagues, who showed that a higher molecular weight is related to a higher affinity between fucoidan and the S-protein of SARS-CoV-2. In an in silico study, a higher molecular weight ligand formed tighter binding to the angiotensin-converting enzyme 2 (ACE2) receptor (which is commonly infected by SARS-CoV-2) [120].

Furthermore, the post-isolation treatment of sulfated polysaccharides might be proposed towards their sulfation degree. A previous study suggests that the desulfation of heparan sulfate, which provides 3-OH octasaccharide, inhibits only 50% of viral entry compared to the same concentration of 3-*O*-sulfated octasaccharide, which completely inhibits viral entry. This study also showed that 3-*O*-sulfated octasaccharide can block membrane fusion by saturating the HSV-1 envelope protein gD in a specific way. This specificity also pointed to the possibility of tuning the sulfation degree of oligosaccharide modification to obtain a specific viral infection inhibitor [121]. The tuning of the sulfation degree can start with comparing commonly isolated carrageenans, which indicates the appropriate degree of sulfation. The sulfate group attached to ι-carrageenan lies between mono-sulfated κ-carrageenan and tri-sulfated λ-carrageenan, which shows superior activity against viral infection.

In addition to the versatile compounds of the purification and post-processing of naturally available sulfated polysaccharides, they also exhibit the ability to interfere with further modification with metals. Interestingly, in 2011, Venkatpurwar and colleagues demonstrated that porphyran can be utilized in the synthesis of gold nanoparticles. The synthesized gold nanoparticles were then loaded with doxorubicin to tune drug release behavior. Furthermore, the doxorubicin-loaded gold nanoparticles enhanced the drug’s cytotoxicity against the human glioma cell line compared to doxorubicin alone. This enhancement might correlate with the increased doxorubicin internalization, which is aided by nanoparticles and their endocytosis ability. Meanwhile, native doxorubicin relies solely on a passive diffusion mechanism [122]. Similar to porphyran, Manivasagan and colleagues utilized a sulfated polysaccharide, namely, fucoidan, to synthesize a stable gold nanoparticle, which was further utilized as a doxorubicin vehicle. The vehicle inhibited the MDA-MB-231 cell line by 5 µg/mL of IC_50_ compared with native doxorubicin by 15 µg/mL [123]. Another finding suggests that fucoidan-gold nanoparticles alone initiate a decrease in cell viability similar to that resulting from cyclophosphamide, a chemotherapy agent. The observed cell viability is at less than 50% at a 25µg concentration, both for fucoidan-gold nanoparticles and cyclophosphamide [124]. Meanwhile, the antioxidative capability of the sulfated polysaccharide–iron complex is superior to that of native sulfated polysaccharides [125]. The sulfated polysaccharide could also increase antibacterial activity by using it as a platform for zinc complexion [126] while providing minimal attention to the relationship of the metal modification of sulfated polysaccharides in antiviral activities.

A few features of the metal modification of sulfated polysaccharides still need to be addressed:The encapsulation amount, which correlates with the efficacy concentration;Consideration of the drug preparation (e.g., injections, solids, or transdermal application);Their preservation capabilities, alongside stabilities both before and after uptake;The impact of the bioavailability of the released drug on systemic circulation;The possibility of controlled released tuning in antiviral delivering agents [127].

The superiority of fucoidan, porphyran, and other sulfated polysaccharides suggests that the backbone of sulfated polysaccharides can interfere with membrane fusion and the performance of the delivery mechanism. An evaluation of slight modifications to the known delivering vehicle on their degree of sulfation, molecular weight, and similarity with the successor path of ι-carrageenan could provide potential antiviral candidates and indicates the specifications needed to overcome viral infections. Owing to their embedded antiviral activities and antioxidative properties, sulfated polysaccharides possess great potential as antiviral agents and delivery vehicles. This potential, combined with the stability exhibited by sulfated polysaccharide nanoparticles and their ability to be controlled within the drug release mechanism, further supports their candidacy as therapeutic agents despite the fact that their study is limited regarding the specific relationship. However, a multidisciplinary approach still needs to be integrated into the study on obtaining, processing, and utilizing sulfated polysaccharides. The developed industry on obtaining sulfated polysaccharides may be used as a benchmark to further utilize sulfated polysaccharides as antiviral agents or delivery vehicles for antiviral prevention and treatment.

## Figures and Tables

**Figure 1 molecules-27-01178-f001:**
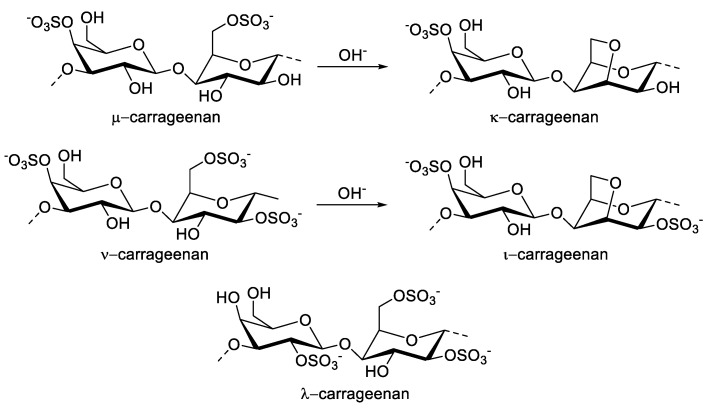
Chemical structure of µ-, κ-, ν-, ι-, and λ-carrageenan with the transformation of carrageenan precursors into κ- and ι-carrageenan under alkaline conditions.

**Figure 2 molecules-27-01178-f002:**
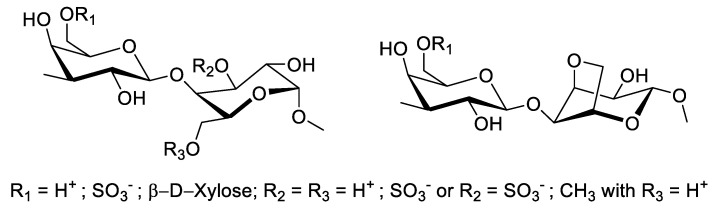
Chemical structure of agaran from *Gymnogongrus tenuis* [57].

**Figure 3 molecules-27-01178-f003:**
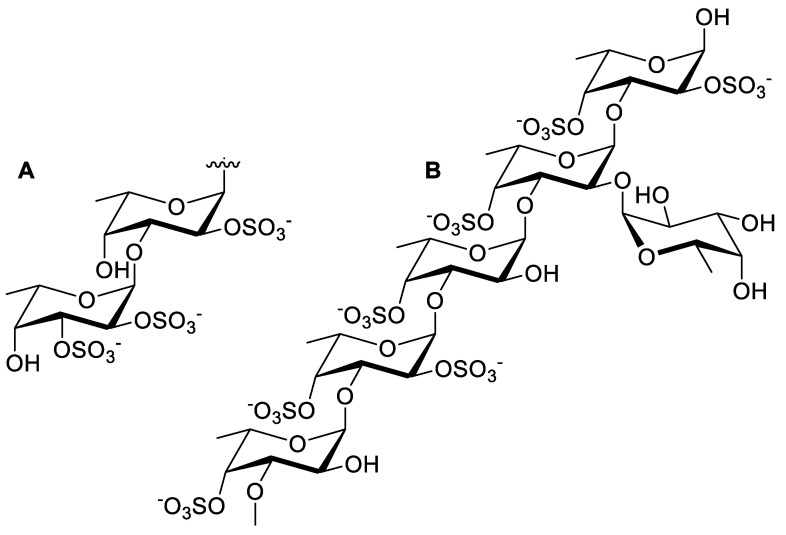
Chemical structure of the fucoidan from (**A**) *Fucus vesiculosus* and *Ascophyllum nodosum* and (**B**) *Laminaria saccharina* [70].

**Figure 4 molecules-27-01178-f004:**
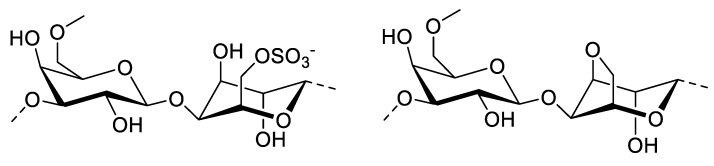
Chemical structure of the typical porphyran.

**Figure 5 molecules-27-01178-f005:**
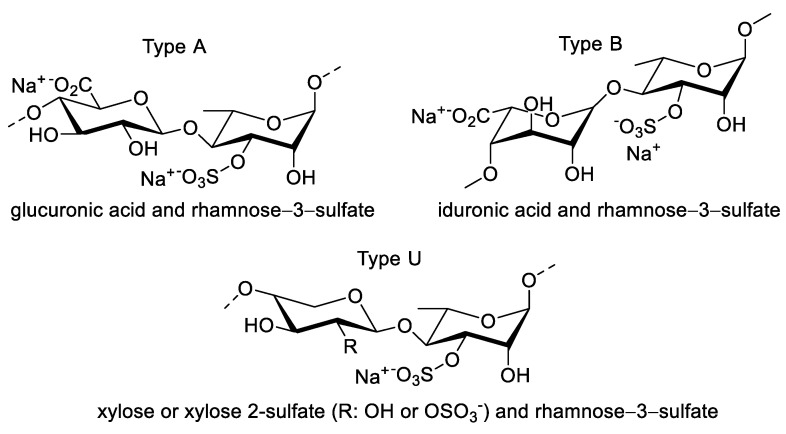
Chemical structure of a typical ulvan [78].

**Figure 6 molecules-27-01178-f006:**
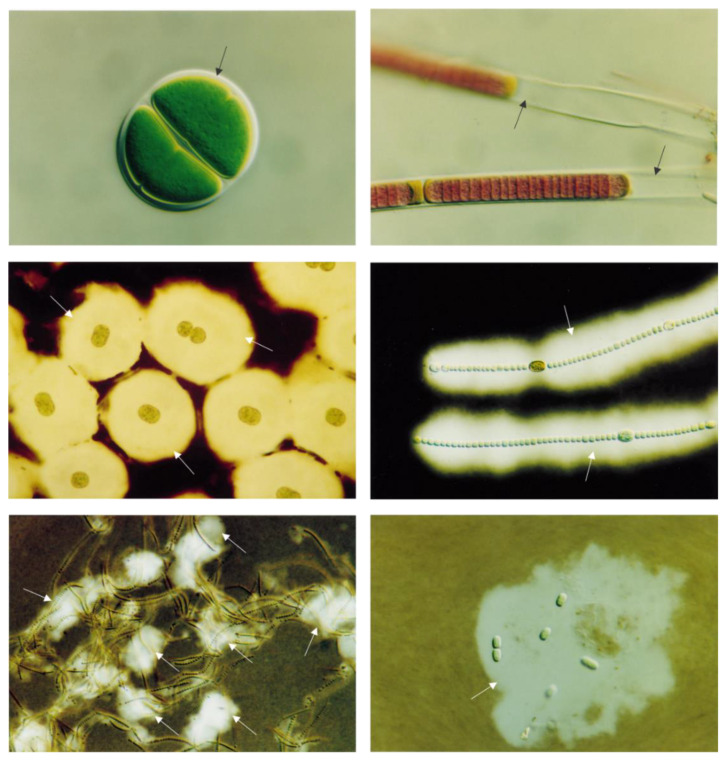
Types of exopolysaccharides (indicated by black and white arrows); top row—thin and dense layer named sheath structure (left—*Chroococcus* sp. (1000×); right—*Phormidium* sp. (1000×)); middle row—the thick layer resembles a cell shape named capsules (left—*Cyanothece* CE 4 (775×); right—*Nostoc* sp. (480×); bottom row—thick layer without cell shape resemblance named slime (left—*Cyanothece* PCC9224 (775×); right—*Nostoc* PCC7906 (194×). Reprinted from De Philippis, R.; Vincenzini, M. Exocellular polysaccharides from cyanobacteria and their possible applications. FEMS Microbiol. Rev. 1998, 22, 151–175, doi:10.1111/j.1574-6976.1998.tb00365.x, with permission from Elsevier.

**Figure 7 molecules-27-01178-f007:**
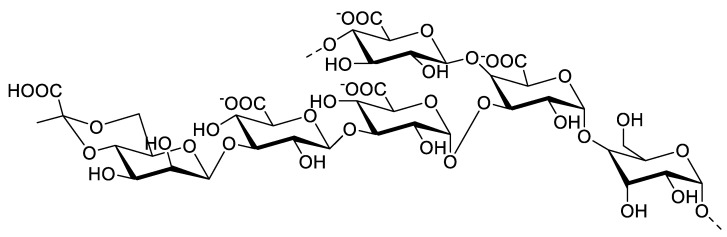
Monomeric unit of exopolysaccharides obtained from the *Alteromonas macleodii* subspecies *fijiensis* [81].

**Figure 8 molecules-27-01178-f008:**
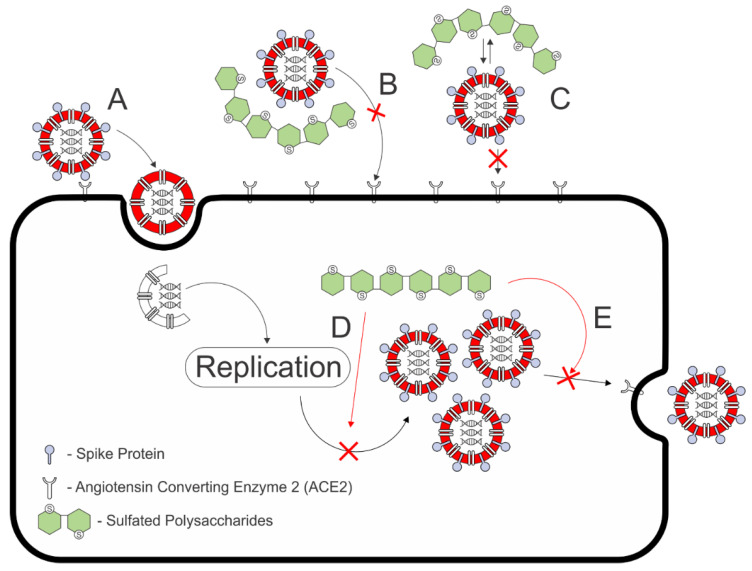
Proposed mechanism of action against viral infection on (**A**) the virion entry; (**B**) inhibition of spike protein binding to ACE2 by complex formation of sulfated polysaccharides-ACE2, (**C**) ionic interaction between the negatively charged sulfate group and spike protein inhibit the formation of spike protein binding to ACE2, (**D**) sulfated polysaccharides inhibition on viral particle replication, indicated by the red arrow, (**E**) inhibition on neuraminidase to prevent assembled viral particle to be released, indicated by red arrow. Mechanisms B and C are proposed for the inhibition against SARS-CoV-2 by [84,88,91]. Mechanisms D and E are proposed from the finding on NDV by [92] and influenza A virus by [12]. Adapted from (Hans, et. al., 2021) and (Frediansyah, 2021).

**Table 1 molecules-27-01178-t001:** Chemical composition of various macroalgae containing sulfated polysaccharides.

Algal Strain	Carbohydrate (%)	Protein (%)	Lipid (%)	Ash (%)	Refs.
Red Algae					
*Gracilaria birdiae*	73.01	8.03	0.46	6.05	[35]
*Kappaphycus alvarezzi*	27.4	16.24	0.74	19.7	[36]
*Mastocarpus stellatus*	35.08	9.14	4.63		[37]
*Porphyra tenera*	46.0	n.d. ^1^			[18]
*Gelidium amansii * ^2^	79.24–86.78	2.22–3.46			[38]
Brown Algae					
*Durvillaea antarctica*	54.57	10.79	0.43	26.06	[39]
*Macrocystis pyrifera*	52.71	9.81	0.21	30.47	
*Lessonia nigrescens*	48.36	9.88	0.23	33.03	
*Sargassum thunbergii*	37.00	7.14	7.88		[37]
*Laminaria japonica*	54.6	8.7			[18]
*Hizikia fusiforme*	94.4	10.9			
*Sargassum horneri*	99.1	4.0			
*Undaria pinnatifida*	60.3	2.6			
*Fucus vesiculosus*		12.99	3.75	20.71	[40]
Green Algae					
*Ulva* sp.	55.40	4.24	6.67		[37]
*Codium fragile*	29.0	1.4			[18]
*Caulerpa veravelensis*	37.23	7.77	2.80	33.70	[41]
*Caulerpa scalpelliformis*	38.84	10.50	3.06	40.77	
*Caulerpa racemosa*	48.95	12.88	2.64	24.20	

^1^ n.d. = not detected; ^2^ The value range represented on the *Gelidium amansii* was taken from three different locations.

**Table 2 molecules-27-01178-t002:** Chemical composition of various microalgae containing sulfated polysaccharides.

Algal Strain	Carbohydrate (%)	Protein (%)	Lipid (%)	Ash (%)	Refs.
Rhodophyta					
*Porphyridium cruentum*	42.17	19.57	5.69	23.59	[52]
*Porphyridium purpureum*	43.88	15.08	1.73	18.57	[53]
Haptophyta					
*Isochrysis galbana*	17.67	28.98	31.09	15.16	[52]
*Ruttnera lamellosa*	63.69	8.81	2.68	43.69	[53]
Cyanobacteria					
*Spirulina platensis*	13.60	56.79	8.33	10.05	[54]
Chlorophyta					
*Tetraselmis suecica*	24.01	26.05	14.68	17.99	[52]
*Chlorella protothecoides*	31.6	48.2	6.9		[55]
*Chlorella vulgaris*	0.47 *	0.90 *	0.36 *		[54]
Bacillariophyceae (Diatom)					
*Phaeodactylum tricornutum*	16.91	26.95	12.73	27.95	[52]
Eustigmatophytes					
*Nannochloropsis oceanica* (post-optimization)	1.0 *	7.58 *	18.25 *		[50]
*Nannochloropsis oceanica* (pre-optimization)	0.69 *	6.40 *	16.43 *		
*Nannochloropsis gaditana*	15.90	33.17	27.89	11.52	[52]

* Values are recalculated from original references to obtain a consistent percent value.

**Table 4 molecules-27-01178-t004:** Summary of antiviral activity among various fucoidan, with each respective proposed mechanism of action and remarks on molecular weight and sulfate content of the tested compound.

Sulfated Polysaccharide	Virus Strain	Antiviral Activities		Proposed Mechanism of Action	Toxicity (Cell)	Remarks on Molecular Weight and Sulfate Content		Refs.
Fucoidan (*Adenocystis utricularis*)	HSV-1	1.25 µg/mL IC_50_			>1000 µg/mL Vero ^1^	1. Lower molecular weight yields less antiviral activity 2. Higher sulfate content yields greater antiviral activity		[94]
	HSV-2	1.63 µg/mL IC_50_						
		immediate addition ^2^	addition after 1 h infection ^2^					
Fucoidan (*Undaria* *pinnatifida*)	HSV-1	2.5 µg/mL	14 µg/mL	Inhibition of viruses into host cell	>2000 µg/mL Vero cell			[95]
	HSV-2	2.6 µg/mL	5.1 µg/mL		>2000 µg/mL Vero cell			
	Influenza A	1.5 µg/mL	55 µg/mL		>2000 µg/mL MDCK cell			
		plaque formation (IC_50_)	Influenza A neuraminidase (IC_50_)					
Fucoidan (*Kjellmaniella crassifolia*)	Infl. A (H1N1)	30.5 µg/mL	8.8 µg/mL	Inhibition of enzyme related to virus adsorption or release process	~80% cell viability at 1000 µg/mL	563 kDa	30.1% sulfate content	[12]
	Infl. A (H3N2)	6.3 µg/mL						

HSV—herpes simplex virus; Infl. A—influenza A; MDCK—Madin–Darby canine kidney cells; ^1^ Vero cell used is African green monkey kidney cells; ^2^ values are presented by IC_50_ for each treatment.

**Table 5 molecules-27-01178-t005:** Summary of antiviral activity among various agaran and ulvan, with each respective proposed mechanism of action and remarks on molecular weight and sulfate content of the tested compound.

Sulfated Polysaccharide	Virus Strain	Antiviral Activities		Proposed Mechanism of Action	Toxicity (Cell)	Remarks on Molecular Weight and Sulfate Content	Refs.
		**IC_50_ of Plaque Formation ^2^**					
Agaran (*Bostrychia montagnei*)	HSV-1	17–24%	13.1–25.7 µg/mL	multiple sulfate inhibits positive charge binding sites of the viral envelope glycoprotein, which is necessary for virus attachment onto cell surface	>1000 µg/mL Vero ^1^	higher molecular weight and sulfate content correlates with greater antiviral activity	[103,104]
		11.2–16.2%	>50 µg/mL				
	HSV-2	17–24%	12.4–46.2 µg/mL				
		11.2–16.2%	>50 µg/mL				
Agaran (*Acanthophora* *spicifera*)	HSV-1	15.1–26.4%	0.6–0.8 µg/mL				
		4.7–6.9%	>50 µg/mL				
	HSV-2	15.1–26.4%	0.9–1.4 µg/mL				
		4.7–6.9%	>50 µg/mL				
		EAE	IC_50_				
Ulvan (*Ulva armoricana*)	HSV-1	carbohydrases	320.9–373.0 µg/mL		>500 µg/mL		[105]
		proteases	>500 µg/mL				
		inhibition of syncytia formation					
Ulvan (*Ulva clathrata*)	NDV	51.54% at 0.1 µg/mL		inhibition of NDV replication cycle	810 µg/mL Vero ^1^		[92]
		JEV-infection inhibition					
Ulva (*Ulva lactuca*)	JEV	70% at 0.03 µg/mL		inhibition of viruses from entering into cells	no toxicity was observed in tested mice	higher MW yielded better anti-JEV activity	[106]

HSV = herpes simplex virus; NDV = Newcastle disease virus; JEV = Japanese encephalitis virus; MW = molecular weight; EAE = enzyme-assisted extraction.^1^ The Vero cell used is African green monkey kidney cells. ^2^ The presented percentage values are the sulfate content range of the fractionated extract.

**Table 6 molecules-27-01178-t006:** Antiviral activities of various sources of exopolysaccharides.

Virus Strain	Exopolysaccharide Source		
	*C. polykrikoides*		
	Fraction A1 (µg/mL)		Fraction A2 (µg/mL)
Influenza A	1.1		0.45
Influenza B	8.3		7.1
RSV-A (Long)	2.0		3.0
RSV-A (FM-58-8)	3.0		2.3
RSV-B	0.8		0.8
HIV-1	1.7		1.7
Parainfluenza virus type 2	25.3		0.8
HSV-1	4.52		21.6
	*Porphyridium* sp. ^1^	*P. aerugineum * ^1^	*R. reticulata* ^1^
HSV-1	1	100	10
HSV-2	5	200	20
VZV	0.7	100	8
	*P. cruentum*	*C. autotrophica*	*Ellipsoidon* sp.
VHSV	50–60% ^2^	<40% ^2^	50–60% ^2^
ASFV	60–70% ^3^	>100% ^3^	40–50% ^3^
	*P. cruentum* (EC_50_ in µg/mL)		
	MgSO_4_ media ^4^	MgCl_2_ media ^4^	Control
HSV-1	34	38	56
HSV-2	12	20	20
*Vaccinia* virus	12	20	20
*Vesicular stomatitis* virus	20	56	100

^1^ Values are presented as the cytopathic effect protection (CPE50) (µg/mL); ^2^ percent focus of infection with respect to control at 20 µg/mL concentration of extracts; ^3^ at 200 µg/mL concentration of extracts; ^4^ media were added with 104 mM of each magnesium salt.

## Data Availability

Not applicable.

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
