# Peer review of "Antiviral Activities of Algal-Based Sulfated Polysaccharides"

_molecules, 2022, doi:10.3390/molecules27041178_

Round 1

Reviewer 1 Report

The submitted manuscript is an interesting and timely review on sulfated polysaccharides and their effect on the infection of some viruses of medical interest.

In general, the manuscript is well written and contains adequate information according to the title. However, I have some comments.

  1. The values in table 2 (superscripts 1 and 2) could be put in the same concentration. To do this, a note would have to be added under the table to say that a new calculation was made in this work.
  2. Add the meaning of all abbreviations used under the tables.
  3. In section 3.6 it may be convenient to add some examples of polysaccharides present in algae, since only some of the constituents are mentioned, but it would be interesting to know some examples.
  4. Figure 6 does not show the structure of exopolysaccharides, at best their location in different organisms is indicated.
  5. I recommend adding the full names or origin of the cell lines before using their abbreviations. For example, HNep (primary human nasal epithelial cells) can be confused with human neutrophil progenitors.
  6. In the description of figure 7 it is necessary to mention which elements belong to the infection or inhibition of coronaviruses and which to those of influenza viruses, because how it is written is confusing.
  7. In the description of figure 7, the reference Has et al., 2021 is not on the final list.
  8. In the section 4.4. Porphyran: Regarding the studies in HPV, it is necessary to mention that these are studies in transgenic mice.
  9. Add the meaning of the abbreviations used at the bottom of each table.
  10. Page 18, line 460: It says, "table 3" but it is "table 6".
  11. In the last section it should be clear that most of the data presented correspond to in vitro studies and that there are very few studies in animal models and none in the clinical phase.

Author Response

Responses to Reviewers` Comments

Manuscript ID Number: molecules-1559092

Manuscript Title: Antiviral Activities of Algal-based Sulfated Polysaccharides

Below are our responses to the comments of reviewers. We have copied and pasted their comments. The reviewer’s as well as editor comments appear in italic and follow in the same order as written by the referee. Our responses appear in regular font.

Comment:

The values in table 2 (superscripts 1 and 2) could be put in the same concentration. To do this, a note would have to be added under the table to say that a new calculation was made in this work.

Response:

Thank you for your remarks, we already recalculated the value on Chlorella vulgaris, Nannochloropsis oceanica (post-optimization), and Nannochloropsis oceanica (pre-optimization) into percent value. The value on Chlorella vulgaris which originates on per 50 grams extract are adjusted to per 100 grams extract. The value on the post- and pre-optimization extraction of Nannochloropsis oceanica is adjusted to per 100 grams extract by multiplies the value to 100 and converted from mg to gram per 100 grams extract.

Comment:

Add the meaning of all abbreviations used under the tables.

I recommend adding the full names or origin of the cell lines before using their abbreviations. For example, HNep (primary human nasal epithelial cells) can be confused with human neutrophil progenitors.

Response:

Thank you for your remarks. On table 3 we already put the abbreviation list below the table and add a few addition to complete the abbreviation presented including the HNep cell with abbreviation addition on page 10 line 301. The same treatment also applied on table 4 and 5.

Comment:

In section 3.6 it may be convenient to add some examples of polysaccharides present in algae, since only some of the constituents are mentioned, but it would be interesting to know some examples.

Response:

Thank you for your remarks, we already added the figure of monomeric unit from exopolysaccharide obtained from Alteromonas macleodii subspecies fijiensis and adjusted all the figure number and remarks on the main text for figures below the added figure.

Comment:

Figure 6 does not show the structure of exopolysaccharides, at best their location in different organisms is indicated.

Response:

Thank you for your remarks, we acknowledge the choice of the word “structure” is not suitable so we decide to only includes the word types to avoid misunderstanding.

Comment:

In the description of figure 7 it is necessary to mention which elements belong to the infection or inhibition of coronaviruses and which to those of influenza viruses, because how it is written is confusing.

In the description of figure 7, the reference Has et al., 2021 is not on the final list.

Response:

Thank you for your remarks, we have added the remarks on the figure caption with specific target on each proposed mechanism tthat involved in antiviral activities,

“…the mechanism B and C are proposed on the inhibition against SARS-CoV-2 by [85,89,92]; the mechanism D and E are proposed from the finding on NDV by [93] and Influenza A virus by [12]…”

Including the citation of Hans, et al., 2021 that have been  added to the final list.

Comment:

In the section 4.4. Porphyran: Regarding the studies in HPV, it is necessary to mention that these are studies in transgenic mice.

Response:

Thank you for your remarks, we alreaedy added the remarks the study in transgenic mice on page 17 line 470.

Comment:

Page 18, line 460: It says, "table 3" but it is "table 6".

Response:

Thank you for your remarks, we already adjust the table number on page 18.

Comment:

In the last section it should be clear that most of the data presented correspond to in vitro studies and that there are very few studies in animal models and none in the clinical phase.

Response:

Thank you for your remarks, we already added the remarks on the limited clinical trial evidence on the efficacy of the algal sulfated polysaccharides.

Reviewer 2 Report

The paper presents all the characteristics of various molecules. 
But I would like to know if there has already been evidence of efficacy in vitro, if there have been it would be useful to present it
English asks for a correction from a native speaker

Author Response

Responses to Reviewers` Comments

Manuscript ID Number: molecules-1559092

Manuscript Title: Antiviral Activities of Algal-based Sulfated Polysaccharides

Below are our responses to the comments of reviewers. We have copied and pasted their comments. The reviewer’s as well as editor comments appear in italic and follow in the same order as written by the referee. Our responses appear in regular font.

Comment:

The paper presents all the characteristics of various molecules.

But I would like to know if there has already been evidence of efficacy in vitro, if there have been it would be useful to present it

English asks for a correction from a native speaker

Response:

Thank you for your remarks, the in vitro effication of sulfated polysaccharides against viral infection are presented on Table 4, 5, and 6 from various virus strain or viral related mechanism of action with remarks on the SARS-CoV-2 related efficacy are included on Table 3 by the reference,

Morokutti-Kurz, M.; Fröba, M.; Graf, P.; Große, M.; Grassauer, A.; Auth, J.; Schubert, U.; Prieschl-Grassauer, E. Iota-carrageenan neutralizes SARS-CoV-2 and inhibits viral replication in vitro. PLoS One 2021, 16, 1–13, doi:10.1371/journal.pone.0237480

and

Jang, Y.; Shin, H.; Lee, M.K.; Kwon, O.S.; Shin, J.S.; Kim, Y. il; Kim, C.W.; Lee, H.R.; Kim, M. Antiviral activity of lambda-carrageenan against influenza viruses and severe acute respiratory syndrome coronavirus 2. Sci. Rep. 2021, 11, 1–12, doi:10.1038/s41598-020-80896-9.

Reviewer 3 Report

The topic of this manuscript is interesting and fits well the scope of the journal. The reviewer feels it can be accepted after some minor amendments.

(1) These compounds appear to be useful. However, they are unlikely to used as therapeutic agent. So the authors should discuss how they can be used?

(2) The safety profile should be discussed.

(3) Are they biologically degradable? How is the compatability? 

Author Response

Responses to Reviewers` Comments

Manuscript ID Number: molecules-1559092

Manuscript Title: Antiviral Activities of Algal-based Sulfated Polysaccharides

Below are our responses to the comments of reviewers. We have copied and pasted their comments. The reviewer’s as well as editor comments appear in italic and follow in the same order as written by the referee. Our responses appear in regular font.

Comment:

The topic of this manuscript is interesting and fits well the scope of the journal. The reviewer feels it can be accepted after some minor amendments.

(1) These compounds appear to be useful. However, they are unlikely to used as therapeutic agent. So the authors should discuss how they can be used?

Response:

Thank you for your remarks, the usage of algal-based sulfated polysaccharides have been discussed on Section 5 Outlook and Future Prospective. We also have been including the safety written by the phrase,

“… Herein, the use of carrageenans is more attractive since most of them have been recognized by Food and Drug Administration, USA as generally recognized as safe (GRAS) product for human consumption (21 CFR 172.620) and supported by low cytotoxicity from other type of sulfated polysaccharides as presented on table 3, 4, and 5. …”

with the addition of CFR number on carrageenan. In addition, iota-carrageenan product as nasal spray also has been recognized for its efficacy on the study by

Figueroa, J.M.; Lombardo, M.E.; Dogliotti, A.; Flynn, L.P.; Giugliano, R.; Simonelli, G.; Valentini, R.; Ramos, A.; Romano, P.; Marcote, M.; et al. Efficacy of a Nasal Spray Containing Iota-Carrageenan in the Postexposure Prophylaxis of COVID-19 in Hospital Personnel Dedicated to Patients Care with COVID-19 Disease. Int. J. Gen. Med. 2021, Volume 14, 6277–6286, doi:10.2147/ijgm.s328486.

Comment:

(2) The safety profile should be discussed.

Response:

Thank you for your remarks, we have discussed the safety matter of the compounds and summarized it on the table 3, 4, and 5, specifically on the Toxicity (Cell) column.

Comment:

(3) Are they biologically degradable? How is the compatability?

Response:

Thank you for your remarks, the natural occurrence of sulfated polysaccharides is followed by the embedded biodegradable capabilities which discussed by,

Ferreira, A.R.V.; Alves, V.D.; Coelhoso, I.M. Polysaccharide-based membranes in food packaging applications. Membranes (Basel). 2016, 6, 1–17, doi:10.3390/membranes6020022.

which also includes the capabilities of carrageenan, one of sulfated polysaccharides, to be used to produce edible films and coatings.

Round 2

Reviewer 2 Report

Accept in this form.